

# qPCR and loop mediated isothermal amplification for rapid detection of *Ustilago tritici*

Hanwen Yan, Jian Zhang, Dongfang Ma and Junliang Yin

Engineering Research Center of Ecology and Agricultural Use of Wetland, Ministry of Education, Hubei Collaborative Innovation Center for Grain Industry, College of Agricultural, Yangtze University, Jingzhou, Hubei, China

## ABSTRACT

Loose smut of wheat caused by the basidiomycete fungus *Ustilago tritici*, a seed-borne disease, is difficult to control because of the expanse of wheat planting area and difficulty in pathogen detection. In this study, real-time fluorescence quantitative PCR (qPCR) and loop-mediated isothermal amplification (LAMP) assays are used to rapidly amplify the DNA of *U. tritici*. Five pairs of primers for qPCR and two series primers for LAMP were designed. Primarily, the specificity of the primer was assessed by using genomic DNA of *U. tritici*, *Fusarium graminearum*, *Blumeria graminis*, *Rhizoctonia cerealis*, *Puccinia striiformis*, *Bipolaris sorokiniana*, and *Alternaria solani* as templates. Further, the amplification systems were optimized. Finally, the sensitivity of qPCR and LAMP assays were evaluated. The results showed that the primer Y-430 F/R, Y-307 F/R, Y-755 F/R, and Y-139 F/R for qPCR and primers L-139 and L-988 for LAMP could be used for *U. tritici* detection. In the sensitivity test, the detection limit of qPCR assay was identified as 10 pg $\mu L^{-1}$ of genomic DNA, the detection limit for LAMP assay was 100 fg $\mu L^{-1}$. We successfully performed qPCR and LAMP assays on wheat loose smut wheat samples. This paper establishes two methods for *U. tritici* detection, which can be used for diagnosis of wheat loose smut in the laboratory and in the field.

## INTRODUCTION

Wheat is one of the most important food crops in the world, and it is also the staple food for most of the world's population (*Garg et al., 2014*). Its global cultivation area is as high as 220,107,600 hectares (*Zhao et al., 2018*). Loose smut of wheat caused by the basidiomycete fungus *Ustilago tritici*, is one of the most serious diseases on wheat (*Triticum aestivum* L.) globally. The disease is favored by moist and cool climate during anthesis (*Quijano et al., 2016*). This fungus converts the spike floral tissues to fungal teliospores, and the carrier seeds of loose smut are the only way of transmission (*Kumar et al., 2018*). The mycelium of *U. tritici* is kept viable in the embryo of the infected seed and moves on the growth point of the tiller without any visible symptoms (*Kumar et al., 2018*), but will be revealed after the heading period of the next year. The yield losses of a single plant are nearly 100% after the onset of disease, and the general disease rate is 1–5%.

Corresponding authors
Dongfang Ma,
madf@yangtzeu.edu.cn
Junliang Yin,
yinjunliang@yangtzeu.edu.cn

When the incidence is serious, it is more than 10%, which can reduce the yield of wheat by 5–20% (*Quijano et al., 2016*). With the popularization of wheat planting area, the damage caused by loose smut in wheat has become more serious, and it has gradually become one of the most harmful diseases in wheat cultivation (*Knox, Menzies & Sharma, 2012*).

At present, the most effective control method of loose smut in wheat is sterilizing before seeding (*Singh et al., 2014b*; *Duan et al., 2016*). However, there are no suitable and effective control measures after sowing. The long-term use of chemical agents can easily pollute the environment, endanger the human health and livestock, and the excessive use of a single drug can easily lead to the emergence of drug resistance (*Crane et al., 2013*). Therefore, timely detection of the pathogen *U. tritici* becomes imperative. Host diversity often leads to diversity of pathogen species, and detection of pathogens requires homozygous strains. Separating and cultivating the strains on the diseased plants is a common effective method for obtaining pure strain. The standard detection method for *U. tritici* is serological identification, but it is time-consuming and needs specific expertise. Meanwhile, its accuracy and sensitivity are not satisfactory (*Walcott, 2003*; *Munkvold, 2009*). Currently, the common detection method for loose smut in wheat is PCR (*Martínez-Espinoza et al., 2003*). However, PCR cannot be used to perform accurate quantitative analysis, and cross-contamination can easily give false positive results (*Bretagne, 2003*). Quantitative PCR (qPCR) has quickly become an indispensable tool in scientific research and clinical diagnosis (*Kuypers et al., 2006*; *Yan et al., 2012*). However, there are some shortcomings, such as the high instrument cost, the need for trained personnel for operation, and not applicable for field testing. *Notomi et al. (2000)* invented a novel method for rapid, efficient, and highly specific amplification of target DNA-loop-mediated isothermal amplification (LAMP). The principle of the experiment is based on design of four different primers (F3, B3, FIP, BIP) for six different positions of the target sequence (F3c, F2c, F1c, B1, B2, B3) (*Tomita et al., 2008*; *Notomi et al., 2015*), acting under the action of *Bst* DNA polymerase, in a water bath instead of the PCR instrument. The reaction is performed at 60–65 °C for 60–90 min and the target DNA amplification is increased to $10^9$–$10^{10}$ (*Dhama et al., 2014*). Compared to PCR, LAMP shortens the reaction time, eliminates the gel electrophoresis step, does not require expensive instruments, and completes the experiment with the *Bst* DNA polymerase in constant temperature conditions. The results are determined by the color reaction of fluorescent dyes. Currently, the fluorescent dyes used are calcein (*Rane et al., 2015*), PicoGreen (*Curtis, Rudolph & Owen, 2008*), hydroxy naphthol blue (HNB) (*Goto et al., 2010*; *Mohon et al., 2014*), and SYBR Green (*Balne et al., 2013*; *Zhou et al., 2014*). SYBR Green I and HNB have the highest detection sensitivity, 10 times that of calcein (*Gao et al., 2009*), and HNB and SYBR Green I can produce long-term stable color changes with brightness in pipes and prevent cross-contamination (*Almasi et al., 2013*). Moreover, their high affinity with double-stranded DNA makes them the most commonly used fluorescent dyes. In this study, SYBR Green I was used in both qPCR and LAMP assay. However, since it binds to any dsDNA molecule, confirming the specificity of the primers is essential for further identification. Therefore, in the qPCR assay, an extra melting curve step was performed to

identify the specificity of the primers. As for the LAMP assay, the primers were screened by PCR to detect any primer dimer formation, and control bacteria DNA was used to determine the specificity of the primers. The combination of LAMP and fluorescent dyes makes bio-detection simpler and more intuitive. At present, LAMP and qPCR have been used for detection of many bacteria, fungi, and viruses and in other microbial detection, such as *Verticillium albo-atrum* (*Tian et al., 2016*), *Listeria monocytogenes* strains (*Wang et al., 2012*), parasites (*Abdul-Ghani, Al-Mekhlafi & Karanis, 2012*), and *Candidatus Liberibacter asiaticus* (*Rigano et al., 2014*). There have been reports on the detection of *Rhizoctonia cerealis* (*Sun et al., 2015a*) and *Tilletia controversa* Kühn (*Nian et al., 2009*) by qPCR assays, and the detection of Fusarium head blight (*Niessen & Vogel, 2010*) and wheat stripe rust (*Huang et al., 2011*) by LAMP assays. However, the detection of wheat loose smut by these two methods has not yet been reported. In this study, we used qPCR and LAMP assays to rapidly detect wheat loose smut (*Kuboki et al., 2003*; *Poon et al., 2005*; *Kono et al., 2004*).

# MATERIALS AND METHODS

## Materials

### Fungal strains

All *U. tritici*, *Fusarium graminearum*, *Blumeria graminis*, *R. cerealis*, *Puccinia striiformis*, *Bipolaris sorokiniana*, *Alternaria solani* strains are provided by the Pathology Laboratory of the College of Agriculture, Yangtze University.

### Culture environment

The wheat variety used in the experiment was Mingxian169 provided by the Pathology Laboratory of the College of Agriculture, Yangtze University. After germination for 24 h in dark conditions, the seeds were planted in a pot and placed in a light incubator at 22 °C for the 12 h of the day and 18 °C for the 12 h of the night. *U. tritici* were collected from the spike tissues of diseased wheat. Wheat powdery mildew was derived from diseased leaves. *F. graminearum*, *Blumeria graminis*, *R. cerealis*, *Bipolaris sorokiniana*, *A. solani* were inoculated on the Potato Dextrose Agar (provided by the Pathology Laboratory of the College of Agriculture, Yangtze University) mediums covered with glass paper and cultured at 25 °C for 7–8 days. *P. striiformis* was collected from the field.

### Genomic DNA extraction

DNA was extracted from wheat loose smut by modified Cetyltrimethylammonium Ammonium Bromide (*Allen et al., 2006*). Briefly, lysis buffer (10 mM Tris-HCl (pH 8.0), 100 mM Ethyl-enediaminetetraacetic acid, 0.5% sodium dodecyll sulfate, and 100 µg mL$^{-1}$ proteinase) was added to the sample, followed by incubation in a water bath at 55 °C for 1 h. The DNA was extracted with by phenol-chloroform-isoamyl alcohol (25:24:1) method, precipitated with isopropanol, and washed with ethanol (70%). After centrifugation, 30 µL of ddH$_2$O was added and the final elute was stored at −20 °C.

**Table 1 Primers designed for qPCR.**

| GenBank | Primer name | Type | Primer sequence (5′–3′) | Production length (nt) |
|---|---|---|---|---|
| JN367334.1 | Y-334 | Forward<br>Reverse | CACGGACCAAGGAGTCTAACAT<br>CCTCTGGCTTCACCCTATTCA | 199 |
| AF135430.1 | Y-430 | Forward<br>Reverse | CCATTTATCGTGGCTCCCTT<br>TACCCATCTCAACCTCTCCG | 134 |
| JN367307.1 | Y-307 | Forward<br>Reverse | CATTTATCGTGGCTCCCTT<br>TCCTACCCATCTCAACCTCTCC | 138 |
| KP256755.1 | Y-755 | Forward<br>Reverse | CTGCTTCTAACAATGCTGACG<br>CAACCATCTTACCTAACCCGC | 162 |
| AJ236139.1 | Y-139 | Forward<br>Reverse | GGGTAGGAGGTCAGAGATGC<br>CGTAAAGGTGCCCGAAGG | 211 |

**Table 2 Primers designed for LAMP.**

| GenBank | Primer name | Type | Sequence (5′–3′) | Length (nt) |
|---|---|---|---|---|
| AJ236139.1 | L-139 | F3 | GGGTAGGAGGTCAGAGATGC | 20 |
| | | B3 | CGTAAAGGTGCCCGAAGG | 18 |
| | | FIP (F1c+F2) | CCGACGTTGGCCTGCAATCT-<br>GGTCTGGGATTCAGCCTTG | 39 |
| | | BIP (B1c+B2) | GTGGAAGGAATGTGGCACCTCT-<br>AGTACGCTGCTGTCCTCG | 40 |
| DQ132988.1 | L-988 | F3 | AAGGGAGCCACGATAAATGG | 20 |
| | | B3 | GGCAACGGATCTCTTGGTT | 19 |
| | | FIP (F1c+F2) | CCTGTTTGAGGGCCGCGAATT-<br>CCGATCCGTCAACCTTTTCC | 41 |
| | | BIP (B1c+B2) | GAGCGCAAGGTGCGTTCAAAG-<br>CGATGAAGAACGCAGCGAA | 40 |

## Methods

### The qPCR and LAMP primer design

Primers were designed by Primer Premier 5.0 based on the sequences of *U. tritici* published in National Center for Biotechnology Information (NCBI). JN367334.1, AF135430.1 (*Bakkeren, Kronstad & Lévesque, 2000*), JN367307.1, KP256755.1 (*Hemetsberger et al., 2015*), and AJ236139.1 were selected for qPCR assay (Table 1). After selecting a large number of DNA sequences for loose smut from NCBI, the sequences were adopted by the multiple sequence alignment to identify highly homologous sequences such as AJ236139.1 and DQ132988.1 and designed primers for AJ236139.1 and DQ132988.1 by Primer Explorer V5 (http://primerexplorer.jp/lampv5e/index.html) for LAMP assay (Table 2). The length of the DNA used was less than 300 bp, and the six parts of the primer were amplified separately for different sequences. Each set of primers consists of two outer primers (F3/B3) and two inner primers (FIP/BIP), FIP containing F1c and F2, BIP containing B1c and B2 (Fig. 1). The primers were synthesized by the Beijing Genomics Institute (Beijing, China), dissolved in ddH$_2$O, and stored at −20 °C.
A

```
                        F3                                 F2
348  5'-GGGTAGGAGGTCAGAGATGCGGTCTGGGATTCAGCCTTGCTTTTGCTGGGTGTTTT-3'  403
                    F1                                    B1c
404  5'-TCTCAGATTGCAGGCCAACGTCGGTTTTGGGCGCTGGAGAAGGGTGGAAGGAATGT-3'  459
460  5'-GGCACCTCTCGGGGTGTGTTATAGCCTTCTACTGGATACAGTGACCGAGACCGAGG-3'  515
         B2c                                  B3c
516  5'-ACAGCAGCGTACTCGCAAGAGCGGGCCTTCGGGCACCTTTACG-3'              558
```

B

```
                        F3                                 F2
133  5'-AAGGGAGCCACGATAAATGGCAAAAACCCTCAATACCGATCCGTCAACCTTTTCCA-3'  188
                      F1
189  5'-AAAGAAAAAAGCTGTCGTTCGAAACAATTCGCGGCCCTCAAACAGGCATGCTCCCC-3'  244
                              B1c
245  5'-GATTAGATCTGCCGGGAGCGCAAGGTGCGTTCAAAGATTCGATGATTCACTTCTG-3'   300
                   B2c                        B3c
301  5'-CAATTCACATTACTTATCGCAATTCGCTGCGTTCTTCATCGATGGGAGAACCAAGA-3'  356
357  5'-GATCCGTTGCC-3'                                               367
```

**Figure 1 Distribution of primers L-139 and L-988 for LAMP assays on DNA sequences.** (A) The position of L-139 in AJ236139.1. (B) The position of L-988 in DQ132988.1.

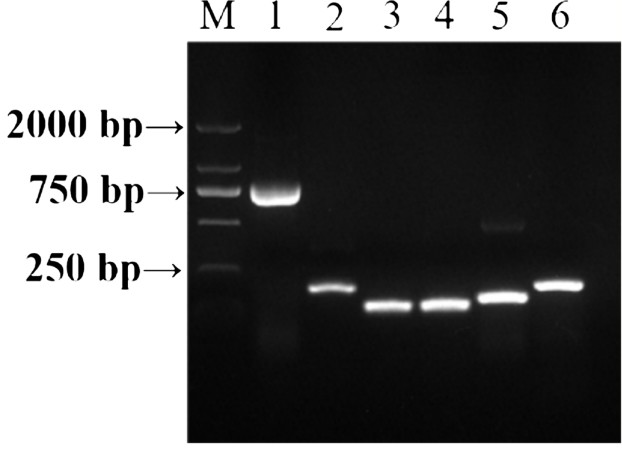

**Figure 2 PCR amplification of template DNA with different primers.** M: DNA maker, 1–6: ITS-4/ITS-5, JN367334.1, AF135430.1, JN367307.1, KP256755.1, AJ236139.1.

## qPCR assays

### Specificity of qPCR assays

DNA of *U. tritici* was used to detect the specificity of the primers. The amplification conditions were initial denaturation at 95 °C for 3 min, followed by 34 cycles of denaturation at 95 °C for 30 s, annealing at 52 °C for 30 s, amplification at 72 °C for 40 s and a final amplification at 72 °C for 5 min. The amplification products were preserved at 16 °C (PCR Thermal Cyclers, Shanghai, China). The amplification products were detected by 1% agarose gel electrophoresis followed by sequencing (Fig. 2). Based on the preliminary screening results, primers were further screened by qPCR using the control strains as templates, and primer specificity was determined by the Ct value presented by the amplification curves (Fig. 3).
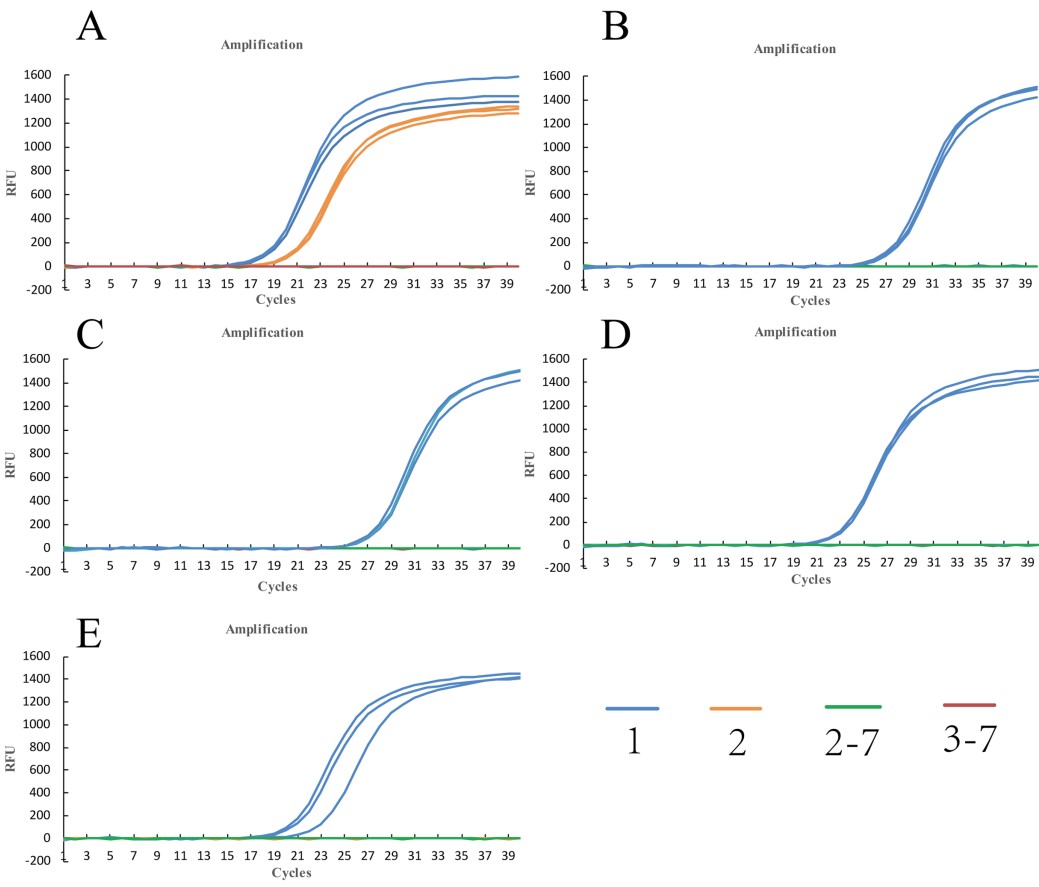

**Figure 3  Amplification curves of qPCR for control fungi using five sets of primers.** (A) primer Y-334. (B) primer Y-430. (C) primer Y-307. (D) primer Y-755. (E) primer Y-139. Color 1: *U. tritici*. Color 2: *F. graminearum*. Color 2–7: *F. graminearum*, *Blumeria graminis*, *R. cerealis*, *P. striiformis*, *Bipolaris sorokiniana*, *A. solani*. Color 3–7: *Blumeria graminis*, *R. cerealis*, *P. striiformis*, *Bipolaris sorokiniana*, *A. solani*.                                

## Optimization of qPCR assays

Appropriate proportion of reaction contents affect the accuracy of the qPCR results, therefore, system optimization becomes indispensable. System optimization experiments were performed using gradients of different volumes of ChamQ$^{TM}$SYBR$^®$ qPCR Master Mix, such as four, six, eight, 10, 12, 14, and 16 μL in qPCR assays (Fig. 4). The ideal temperature was determined after demonstrating the optimum system proportions. Seven temperature gradients were designed, 52, 54, 56, 58, 60, 62, and 64 °C (Fig. 5). Melting temperature and gel electrophoresis were combined to determine the optimum temperature.

## qPCR for U. tritici

For a total reaction volume of 20 μL, the reaction mix comprised the following one μL DNA, one μL of individual forward and reverse primer, 10 μL Master Mix, and seven μL ddH$_2$O. The reaction conditions were initial denaturation at 94 °C for 3 min followed by 40 cycles of denaturation at 94 °C for 20 s, primer annealing at 60 °C for 30 s, and

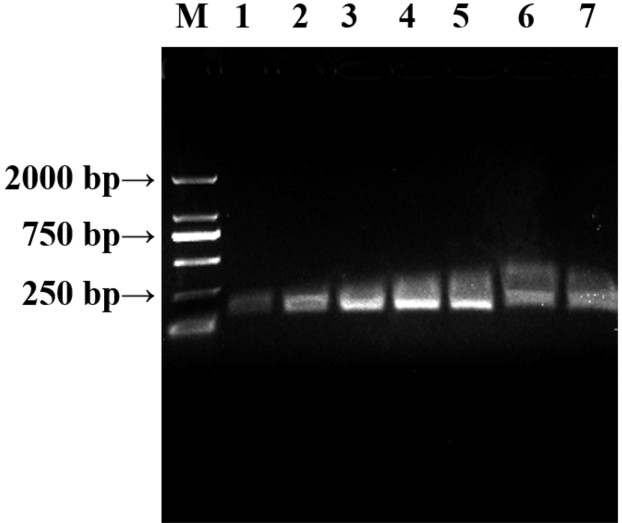

**Figure 4 Volume optimization of qPCR for ChamQ™SYBR® qPCR master mix.** M: Maker, 1–7: four, six, eight, 10, 12, 14, 16 μL.

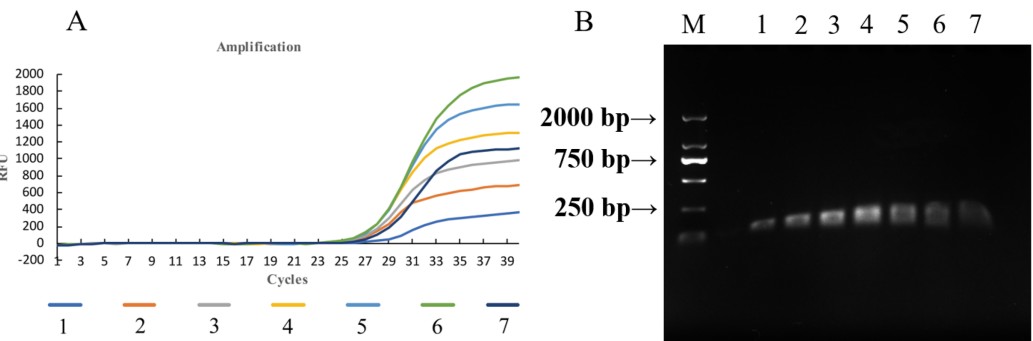

**Figure 5 Temperature optimization of qPCR.** (A) Amplification curves of qPCR for *U. tritici* under temperature gradients. 1–7: 52, 54, 56, 58, 60, 62, 64 °C. (B) Gel electrophoresis of qPCR on *U. tritici* under temperature gradients M: Maker, 1–7: 52, 54, 56, 58, 60, 62, 64 °C.

amplification at 72 °C for 5 min. Each template DNA was diluted to six concentration gradients, with triplicates in each group. We then analyzed melting curves to verify the specificity of the results (Fig. 6).

### Sensitivity detection of qPCR assays

A total of 100 ng μL$^{-1}$ DNA of *U. tritici* was used to the template (*Singh et al., 2014a*) and diluted to seven concentration gradients, 100 ng μL$^{-1}$, 10 ng μL$^{-1}$, one ng μL$^{-1}$, 100 pg μL$^{-1}$, 10 pg μL$^{-1}$, one pg μL$^{-1}$, 100 fg μL$^{-1}$. We performed qPCR on Y-430 with two replicates per concentration (Fig. 7).

## LAMP assay
### Specificity of LAMP assays

We analyzed whether the primers were normal and whether there was primer–dimer formation in PCR. The PCR system for a volume of 25 μL was as follows: DNA 0.5 μL,

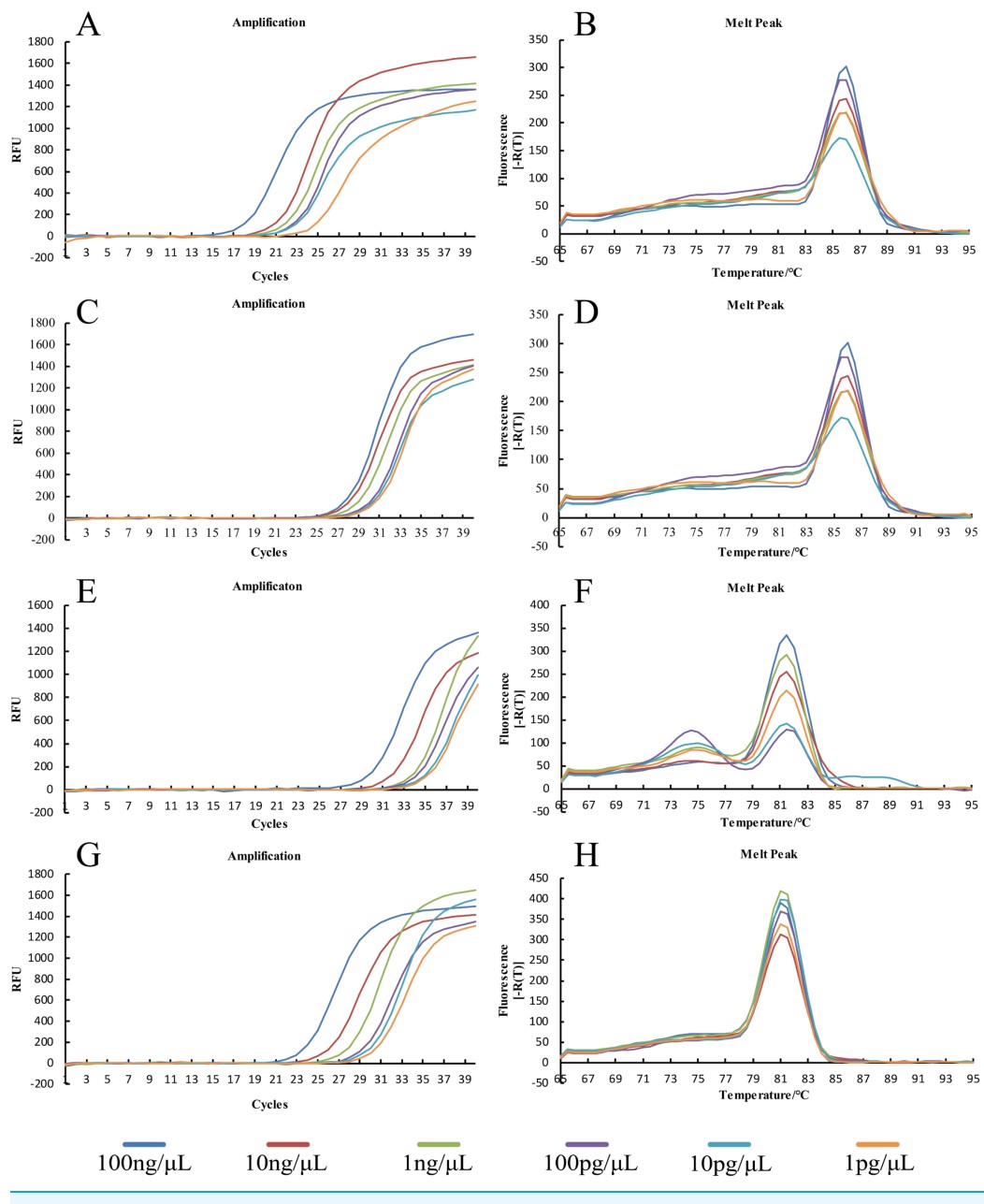

**Figure 6 qPCR for *U. tritici*.** The amplification curves of qPCR for AJ236139.1 (A), KP256755.1 (C), JN367307.1 (E), AF135430.1 (G). The melting curves of qPCR for AJ236139.1 (B), KP256755.1 (D), JN367307.1 (F), and AF135430.1 (H).               

dNTPs one µL, *Taq* DNA polymerase one µL, F3 one µL, B3 one µL, $MgSO_4$ one µL, 2 × Phanta Max Buffer 12.5 µL, $ddH_2O$ seven µL, the two sets of primers F3-1F/B3-1R, F3-2F/B3-2R, with three replicates for each set, $ddH_2O$ was used as control. Fluorescent dye was added to observe the amplification results (Fig. 8). Additionally, *F. graminearum*, *Blumeria graminis*, *R. cerealis*, *P. striiformis*, Bipolaris *sorokiniana*, and *A. solani* were used as the controls for LAMP assay (Fig. 9).
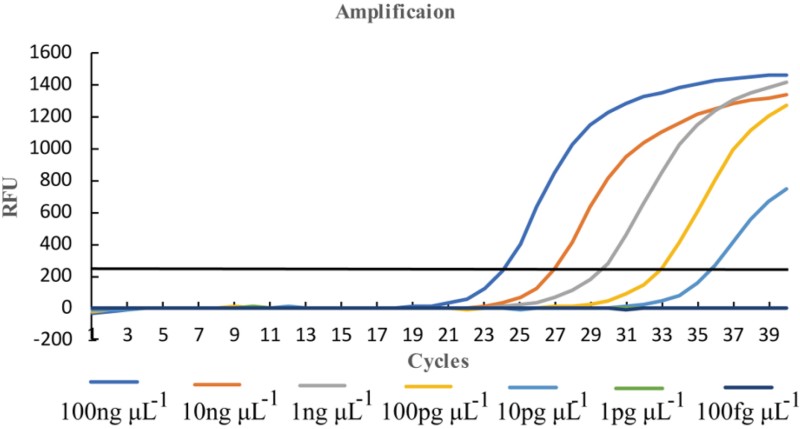

**Figure 7 Sensitivity detection of qPCR assays by primer Y-430.**

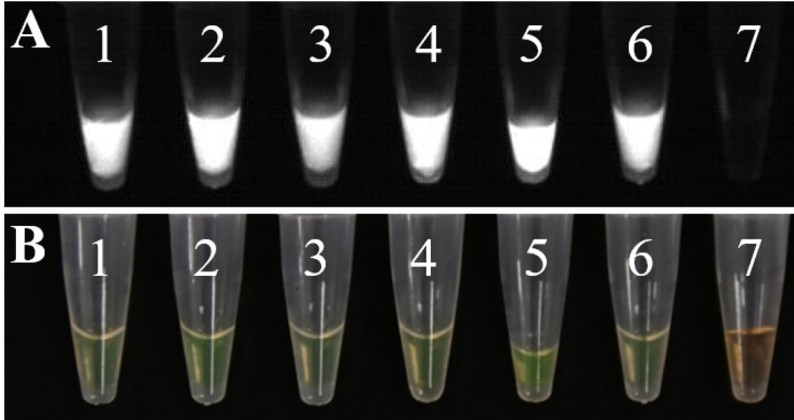

**Figure 8 The results of PCR.** (A) Observing the amplification results in the gel imaging system. (B) Observing the amplification results by the naked eye. 1–3: L-139. 4–6: L-988. 7: Negative control.

## Temperature optimization for LAMP assay

In order to determine the accuracy and sensitivity of the results, the system concentration and temperature for the LAMP assay were optimized. The concentration presented here is the optimal concentration ratio. The concentration ratio of the inner and outer primers used in this experiment was 8:1, the concentration of FIP and BIP was 1.6 µmol L$^{-1}$, and of F3 and B3 was 0.2 µmol L$^{-1}$. The concentration of Mg$^{2+}$ after referral to relevant literatures (*Kubota et al., 2008*; *Abd-Elsalam et al., 2011*) was established as six mmol L$^{-1}$. And nine temperature gradients were designed for LAMP to determine the optimum temperature based on the final color of the reaction (Fig. 10A).

## Sensitivity detection of LAMP assays

We diluted the extracted DNA by 10-fold and then used it as the template for further diluting the template DNA to obtain nine concentration gradients, which were 100 ng µL$^{-1}$, 10 ng µL$^{-1}$, one ng µL$^{-1}$, 100 pg µL$^{-1}$, 10 pg µL$^{-1}$, one pg µL$^{-1}$, 100 fg µL$^{-1}$,

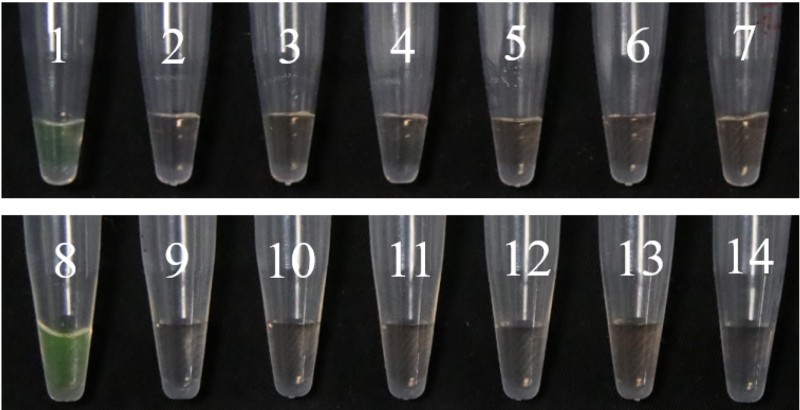

**Figure 9 Specific detection of LAMP assays.** 1–7: The results of LAMP assay with L-139. DNA from 1 to 7: *U. tritici*, *F. graminearum*, *Blumeria graminis*, *R. cerealis*, *P. striiformis*, *Bipolaris sorokiniana*, *A. solani*. 8–14: The results of LAMP assay with L-988. DNA from 8 to 14: *U. tritici*, *F. graminearum*, *Blumeria graminis*, *R. cerealis*, *P. striiformis*, *Bipolaris sorokiniana*, *A. solani*.

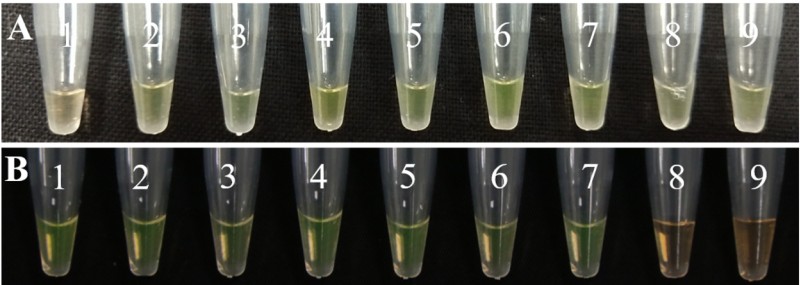

**Figure 10 Temperature optimization and sensitivity detection of LAMP assays.** (A) 1–9: 58, 59, 60, 61, 62, 63, 64, 65, 66 °C. (B) DNA concentration from one to 9: 100 ng uL$^{-1}$, 10 ng uL$^{-1}$, one ng uL$^{-1}$, 100 pg uL$^{-1}$, 10 pg uL$^{-1}$, one pg uL$^{-1}$, 100 fg uL$^{-1}$, 10 fg uL$^{-1}$, one fg uL$^{-1}$.

10 fg $\mu$L$^{-1}$, and one fg $\mu$L$^{-1}$. The results were analyzed by observation under natural light and differentiated by color of the reaction (Fig. 10B).

### LAMP assay for U. tritici

The final volumes of the components of the LAMP reaction system of 25 $\mu$L used in the experiment were as follows: one $\mu$L DNA, 2.5 $\mu$L 10 × *Thermopol* buffer, two $\mu$L MgSO$_4$, two $\mu$L dNTPs, one $\mu$L F3, one $\mu$L B3, one $\mu$L FIP, one $\mu$L BIP, one $\mu$L *Bst* DNA polymerase, five $\mu$L betaine, and 7.5 $\mu$L ddH$_2$O, with triplicates for each set of primers. ddH$_2$O was used as a template in control reactions (Fig. 11).

## RESULTS

In order to determine the specificity of the designed primers, we performed PCR experiments on the DNA of the *U. tritici* (Fig. 2). Accordingly, five sets of sequence bands corresponding to the primers were obtained. Based on the preliminary screening results, primers were further screened by qPCR using the control strains as templates, and

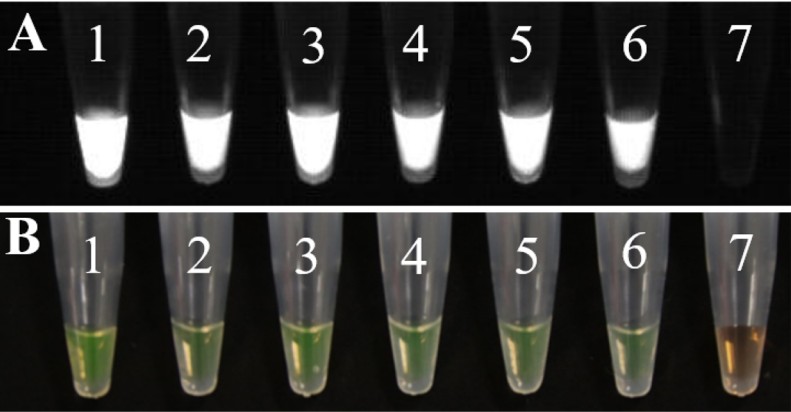

**Figure 11 The results of LAMP assay.** (A) Observing the amplification results in the gel imaging system. (B) Observing the amplification results of the LAMP assays by the naked eye. 1–3: L-139. 4–6: L-988. 7: Negative control.

specificity was determined by the Ct value of the amplification curves (Fig. 3). The results showed that Y-334 amplified *F. graminearum* (Fig. 3A) and therefore it was non-specific for *U. tritici*. The primers Y-430, Y-307, Y-755, and Y-139 specifically amplified the target DNA. Four sets of primers were used to detect *U. tritici*. By optimizing the system, the optimal volume for ChamQ™SYBR®qPCR Master Mix could be formulated. The results of gel electrophoresis after amplification are shown in Fig. 3. The amplification results were the best for a master mix volume of 8–12 µL. We considered the intermediate value is 10 µL, as the optimal volume for ChamQ™SYBR® qPCR Master Mix in qPCR assay. Additionally, the results of qPCR temperature gradient test in *U. tritici*, revealed that the relative fluorescence units reached a high detectable levels at 58 and 60 °C, and the Ct value was about 33, which was in line with our expected results. The bands at 58 °C were the brightest as shown by gel electrophoresis. In accordance with these results (Figs. 5A and 5B), we chosen 58 °C as the temperature for the final experiment. The amplification curves for the four pairs of primers show that the Ct values ranged from 29 to 35 in the samples (Fig. 6). It indicated that the four sets of primers could amplify the target DNA under certain concentration conditions. The melting curves corresponding to each primer exhibited a single peak, which further ruled out non-specific amplification. Combining the amplification curves and the melting curves, it can be stated that the target DNA can be successfully amplified by using the four pairs of primers designed to quantify the *U. tritici*. Finally, we used 100 ng µL$^{-1}$ DNA by diluting it to give seven concentration gradients for verification of lowest concentration detectable by qPCR. The fluorescence results of seven gradients showed (Fig. 7) that the first five gradients gave high signal noise so that the lowest detectable DNA concentration by qPCR was 10 pg µL$^{-1}$.

In LAMP assays, the results of the PCR system analyzed under natural light and gel imager (Fig. 8) indicated that primers designed for LAMP assay can be used for PCR amplification. The pathogens such as *F. graminearum* (Table 3) were used as controls to verify the specificity of the primers. Additionally, five other fungal diseases commonly found in wheat were selected as controls. At the same time, a group of non-wheat fungi

**Table 3 Experimental and control strains.**

| Species | No. of strains | Host plants | PCR | | LAMP | |
|---|---|---|---|---|---|---|
| | | | L-139 | L-988 | L-139 | L-988 |
| *Ustilago tritici* | 2 | Wheat | + | + | + | + |
| *Fusarium graminearum* | 2 | Wheat | – | – | – | – |
| *Blumeria graminis* | 2 | Wheat | – | – | – | – |
| *Rhizoctonia cerealis* | 3 | Wheat | – | – | – | – |
| *Puccinia striiformis* | 3 | Wheat | – | – | – | – |
| *Bipolaris sorokiniana* | 3 | Wheat | – | – | – | – |
| *Alternaria solani* | 1 | Tomato | – | – | – | – |

Note:
"+" means the amplification result is positive, "–" means the amplification result is negative.

were also selected. L-139 and L-988 were used to perform LAMP assays on seven different bacteria (Fig. 9). The results showed that L-139 and L-988 only amplified the DNA sequence of *U. tritici*. As can be seen from Fig. 10A, the optimum reaction temperature for LAMP assay was from 62 to 64 °C. This experiment was performed at 63 °C. The LAMP assay was performed under the optimal reaction proportions and temperature, for *U. tritici*. Through the gel imaging system (Fig. 11A), the positive samples were white and the negative were colorless. Under natural light (Fig. 11B), the three replicates of the two sets of primers were bright green, and the negative controls were light orange. The light orange color of the negative controls indicated the absence of primer-dimers and false positives due to external contamination. Sensitivity testing of the LAMP test indicated that the DNA concentration of the lowest *U. tritici* detectable by the LAMP is 100 fg $\mu L^{-1}$. We performed multiple verifications for experimental accuracy in order to test seed carriers and compare the two methods in terms of sensitivity and operation, so we did not distinguish between the different species.

Based on the successful amplification of DNA from *U. tritici* by qPCR and LAMP assays, Y-430 was used to perform qPCR and L-139 and L-988 were used to perform LAMP assay on diseased seeds (Fig. 12). The qPCR results give a consistent Ct value at 27, and the melting curve also showed a single peak. Similarly, in the LAMP experiment, both samples showed bright green color, and the expected ladder band appeared in agarose gel electrophoresis. Combining the results of the two methods, we conclude that qPCR and LAMP technology can be used for efficient and sensitive detection of *U. tritici*.

## DISCUSSION

Loose smut of wheat is a systemic disease infesting flower organs (*Ngugi & Scherm, 2006*). It currently occurs in all wheat-growing regions of the world, particularly prevalent in Canada (*Randhawa et al., 2009*) and parts of Africa (*Zegeye, Dejene & Ayalew, 2015*). The detection of loose smut in seeds is particularly important due to the increased area of the disease and a single method of prevention. Both qPCR and LAMP assays can specifically and efficiently amplify the DNA of *U. tritici* in this study.

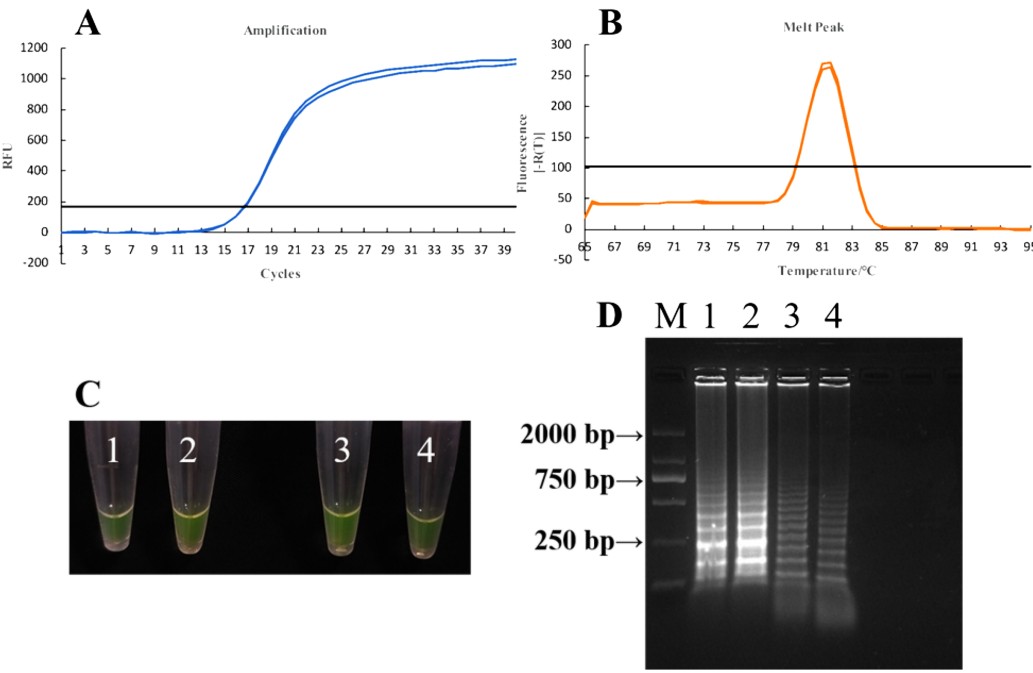

**Figure 12 Detection of diseased seed in the field by qPCR and LAMP assays.** (A) Amplification curves of qPCR for detection of wheat loose smut in field by primer Y-430. (B) Melting curves of qPCR for detection of wheat loose smut in field by primer Y-430. (C) LAMP detection of wheat loose smut in the field under natural light. Tube 1–2: the results of LAMP assay with L-139. Tube 3–4: the results of LAMP assay with L-988. (D) Detection by agarose electrophoresis after LAMP assay. Strip 1–2: the results of LAMP assay with L-139. Strip 3–4: the results of LAMP assay with L-988.

Our ultimate goal was to develop a simple and high-speed detection method. Both the qPCR and LAMP are beneficial to detect pathogens from the source. By comparing the two methods, it is not difficult to conclude that the qPCR assay can accurately determine the initial amount and amplification of the template; it can also be used to visually determine whether there was non-specific amplification by comparison of the melting curves. Combining the amplification curves with melting curves, it was concluded that the primers Y-430, Y-307, Y-755, and Y-139 can specifically, efficiently, and accurately detect *U. tritici*. This method avoids the use of gel electrophoresis and other supplementary operations. Therefore, it has been widely used in the analysis of gene expression (*Ma et al., 2013*), virus detection (*Albinana-Gimenez et al., 2009*), disease diagnosis (*Moreira et al., 2013*). Compared with qPCR, the LAMP assay has the advantage of sensitivity, time, ease of operation, and equipment requirement (*Kiddle et al., 2012*). In combination with fluorescent dyes, the colored reaction is more favorable for observation of the amplification results and can be used widely in biological disease detection (*Jung et al., 2015*), medical diagnosis (*Hopkins et al., 2013*), food testing (*Sun et al., 2015b*), and other aspects. Compared with PCR, both the qPCR and LAMP assays can effectively avoid the influence of agarose gel electrophoresis and the minimum detectable concentration is lower than that by PCR. However, there are still many problems in the application of the two methods. Primarily, the instruments for performing qPCR assays are expensive. The LAMP assay

can only detect one disease at a time. The combination of fluorescent dye with dsDNA is not specific, so the LAMP assay has a high false positive rate. Meanwhile the field environment is complex, the reaction system and concentration ratio are difficult to optimize, etc.

At present, the qPCR and LAMP assays are being improved up. Based on qPCR, multiplex PCR has emerged to compensate for the shortcomings of detecting only one disease at a time. The various conditions of qPCR assay limits its ability for application in field-based assays. Accurate analysis of the data and good reproducibility of the experiment make it the ubiquitous mainstay of molecular biology. With the progress of molecular biology, fluorescence quantification has become an indispensable tool. Meanwhile, multiple LAMP assays have also been proposed to accelerate the efficiency of detection (*Chen et al., 2016*; *Lodh et al., 2017*). Kits for LAMP assays have been developed (*Marti, Stalder & González, 2015*), that eliminate the need for system optimization, thus simplifying the procedure. Its advantages make it suitable for use in resource-poor areas. With advancements in technology and human knowledge, chemical control methods of disease resistance will be gradually replaced by early prevention. The LAMP methodology is a very valuable diagnostic alternative with a potential for use in endemic diseases. The improvements in LAMP technology will make it more effective in disease prevention and control.

## CONCLUSIONS

qPCR and LAMP methods for the detection of *U. tritici* have been developed in this study; both of them have great significance for the diagnosis of *U. tritici*. qPCR has better sensitivity and comparable specificity to current diagnostic tests. LAMP has better availability in laboratories of various standards, and has the potential to be utilized on a large scale.

### Funding

This work was funded by the National Key R and D Program of China (2018YFD0200500), Open Project Program of Key Laboratory of Integrated Pest Management Crops in Central China, Ministry of Agriculture. P. R. China/Hubei Key Laboratory of Crop Disease, Insect Pests and Weeds Control (2017ZTSJJ7). The funders had no role in study design, data collection and analysis, decision to publish, or preparation of the manuscript.

### Grant Disclosures

The following grant information was disclosed by the authors:
National Key R and D Program of China: 2018YFD0200500.
Open Project Program of Key Laboratory of Integrated Pest Management Crops in Central China, Ministry of Agriculture, P. R. China/Hubei Key Laboratory of Crop Disease, Insect Pests and Weeds Control: 2017ZTSJJ7.

## Competing Interests

The authors declare that they have no competing interests.

## Author Contributions

- Hanwen Yan conceived and designed the experiments, performed the experiments, analyzed the data, prepared figures and/or tables, authored or reviewed drafts of the paper, collection and cultivation of experimental strains.
- Jian Zhang conceived and designed the experiments, performed the experiments, authored or reviewed drafts of the paper, fluorescence quantitative PCR operation and data collation.
- Dongfang Ma conceived and designed the experiments, contributed reagents/materials/analysis tools, authored or reviewed drafts of the paper, approved the final draft, modifications, and suggestions for the article.
- Junliang Yin conceived and designed the experiments, contributed reagents/materials/analysis tools, authored or reviewed drafts of the paper, approved the final draft, modifications, and suggestions for the article.

## Data Availability

Data is available at NCBI GEO: AJ236139.1, DQ132988.1, JN367334.1, AF135430.1, JN367307.1, and KP256755.1.

## Supplemental Information

Supplemental information for this article can be found online at http://dx.doi.org/10.7717/peerj.7766#supplemental-information.

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
