# Peer review of "qPCR and loop mediated isothermal amplification for rapid detection of Ustilago tritici"

_PeerJ, doi:10.7717/peerj.7766_

## Round 0.1 · original submission · Major Revisions

The reviews are focused on the language and style, which has to be significantly improved, preferably proofread by an English-editing service or a native English-speaking person.

The technical details of the manuscript are also raised by both reviewers, which include information on how the nested primers for LAMP were designed, tested, and the actual sequence of these primers. Also, access to the supplemental files has to be checked.

Here are some examples of articles published in PeerJ reporting the use and application of the LAMP method:

https://peerj.com/articles/5993/
https://peerj.com/articles/3799/
https://peerj.com/articles/6068/

Reviewer 1 ·

Basic reporting

In this article Yan et al. attempted to describe novel approaches for rapid detection of Ustilago tritici by qPCR and loop mediated isothermal amplification methods. However, obtained data in the current manuscript were a preliminary screening of several factors and trials in order to get an initial outline of how designed primers work in vitro conditions. No description of scientific hypothesis driven to or how to achieve those unique and specific primers was shown in this research. Importantly, no practical experiments were conducted to examine those methods with samples from the fields. Other than that, the use of English in this manuscript was poor and did not meet the professional standard for publishing as a scientific research article. Therefore, I do not think the current version of the article is worth for publication in this journal.

The introduction was unfocused, particularly lacking a description of wheat loose smut’s pathogen, its life cycle, as well as which candidate factors that current approaches are targeted to establish a diagnostic strategy.

The supplemental files of this manuscript were impossible to access. There may be an error from the journal system I would say.

Not only tables of primers (Table 1 as in line 98 and Table 2 as in line 101) were absence, how distinct approaches to design as well as how unique of the designed primers compared to previous publications were also undescribed in this manuscript.

There was no data throughout manuscript representing for the statement in line 27-29 of abstract “We successfully performed qPCR and LAMP assays on two wheat loose smut wheat samples, and confirmed sequenced U. tritici infection by subsequently sequencing”.

Experimental design

More experiments must be conducted to address these key questions:

1. What kind of wheat loose smut samples (soil, wheat stems, seeds, etc.) in fields would be ideal targets for these diagnostic methods?

2. How these methods could be applied to the field samples? Directly running the raw samples or performing with isolated DNA from the samples?

3. It may be a very big question, but let’s think about how many percent or positive obtained results would be the threshold line for a conclusion of a disease outbreak or a need to be controlled?

Validity of the findings

The data obtained in vitro by isolated DNA screening would be generally different from raw field samples. Therefore, based on these preliminary data, additional experiments applied in field samples must be provided to demonstrate the feasibility of the methods.

Additional comments

The manuscript must be revised by not only a senior scientist but also a native speaker to obtain professional criteria of an original research article.

·

Basic reporting

Line 3 & 4: replace “/” with comma
Line 18: Please add seed-borne fungal disease in the first sentence
Line 20: replace “were” with are and “specify” with specificity
Line 21: correct the sentence “were carried out” to is assessed by using genomic DNA of …..
Line 25: Please make the primer nomenclatures consistent throughout the manuscript. Like L-139 vs. L139
Line 29: replace “established” to establishes
Line 30: replace “could” to “can” and diagnose to “diagnosis.”
Line 34-38: Authors did not mention the pathogenesis of the wheat loose smut disease. Two or three sentences for the introduction of the disease and a few sentences for the necessity of the diagnostic test would help the reader get a gist of the manuscript.
Lane 42: incomplete sentence “single used…”
Line 81: replace “bacterial strain” with Fungal strains
Line 83: provide the source of fungi
Line 85: replace “24 h” to 24 hours (h)
Line 86: sentence unclear. Please replace “12h days …..” to for the 12h of the day and 18deg C, 12h of the night.
Line 86 and 87: provide the source of U. tritici and wheat powdery mildew. A supplementary figure with the phenotypical observations of the fungi will be very informative for the readers.
Line 88: Provide the source of PDA media
Line 91: please cite CTAB method and introduce the acronym
Line 99: please cite articles publishing the DNA sequences. Which genome sequences were chosen for the qPCR and LAMP assay? and why? Are there any reference genes or housekeeping genes selected as a baseline for the qPCR data analysis?
Line 103: replace “QPCR” to “qPCR”
Line 105: replace “are……” with is used to detect the specificity of the primers.
Line 105: Provide the details of the thermocycler used.
Line 118: replace “QPCR” to “qPCR”
Line 159: replace “there for” with therefore
Line 192: please confirm qPCR primer Y334 or Y430
Line 205: correct the sentence both……
Figure legends: please change all the fungal species names to the italic font.

Experimental design

Authors approach to develop a specific, rapid, and accurate DNA-based diagnostic test for the wheat loose smut is encouraging. However, the diagnostic test lacks robust experimental data to support the findings and cost-analysis.

Validity of the findings

There is a gap in the explanation of how the detection of an organism can help prevent the disease and why it is so essential to distinguish a species. Authors can discuss these topics in the introduction section. Authors did not explain the strategy in designing the primer set. If the genome sequence of the U. tritici is annotated, then please present and explain the goal behind the amplification of this specific region in the genome. Also, the authors may include a fungal housekeeping gene as a reference gene to confirm relative amplification of the target gene.

---

## Round 0.2 · accepted · Accept

The authors have corrected many grammatical errors and addressed the issues raised by the reviewers. Therefore it can be accepted at this time.